# Changes in urgent and emergency care activity associated with COVID-19 lockdowns in a sub-region in the East of England: Interrupted times series analyses

Zillur Rahman Shabuz, Max Bachmann👤*, Rachel Cullum👤, Amanda Burke👤, Charlotte Emily Louise Jones👤, Oby Otu Enwo👤, Alice M. Dalton, Julii Brainard👤, Michael Saunders👤, Nicholas Steel

Norwich Medical School, University of East Anglia, Norwich Research Park, Norwich, United Kingdom

* m.bachmann@uea.ac.uk

## Abstract

### Background

Access to and use of urgent and emergency care in the United Kingdom's National Health Service reduced during COVID-19 related lockdowns but demand reportedly increased since then. We investigated the impact of COVID-19 on urgent and emergency health care services activity in an eastern England population of 1.1 million.

### Methods

We used health care activity data from a municipal health department, recorded at the level of discrete events (such as visits to hospital or ambulance calls) to compare system activity between 2018–2020 (pre-COVID), 2020–2021 (lockdown) and 2021–2023 (post-lockdown), carrying out interrupted time series analyses to describe changes in activity.

### Results

Daily emergency department (ED) attendances were 10% (95% confidence interval 9–12%) lower during the lockdown period, and 7% (6–8%) higher in the post-lockdown period than pre-COVID. Attendances arriving by ambulance were 13% (12–14%) lower post-lockdown than pre-COVID, while attendances of arrivals by other means were 17% (16–19%) higher. Post-lockdown, overall attendances were continually reducing. ED waiting times were 45% (44–47%) longer in the post-lockdown period compared to the pre-COVID period and continued to increase post-lockdown. There was a 15% (14–16%) reduction in daily ambulance dispatches post-lockdown versus pre-COVID. Ambulance arrivals with delayed handover to hospital care exceeding 60 minutes increased by 17% (16–18%) post-lockdown versus pre-COVID, and probability of delay showed a continuously upward trend post-lockdown of 20% (19–21%) per year.

**Data Availability Statement:** The data are individual record level datasets containing information about each urgent and emergency care

event, including each patient pseudonymised identify number and characteristics of care provided. Data are confidential pseudonymised medical records held by the Department of Public Health, Norfolk County Council, and not by the authors who are not authorised to legally distribute or share these data. Data analysis for the study has been done by the authors under a secure digital environment, and subject to the data governance training and requirements, of Norfolk County Council. Data requests may be sent to: Joint Strategic Needs Assessment, Norfolk County Council, County Hall, Martineau Lane, Norwich, Norfolk, NR1 2DH, email JSNA@norfolk.gov.uk.

**Funding:** This paper presents independent analysis and research supported by Norfolk County Council (https://www.norfolk.gov.uk/). The views expressed are those of the authors and not necessarily those of Norfolk County Council, the National Health Service, or the Department of Health and Social Care. The funders did not have any role in the analysis or interpretation of data or in writing the manuscript. The funders had no role in the decision to publish. NS is the Principal Investigator and was awarded the grant from Norfolk County Council, number R210935.

**Competing interests:** The authors have declared that no competing interests exist.

## Conclusion

Patients are facing long waits in EDs to be admitted to hospital, discharged or transferred. This results in delays in ambulances handing over patients and attending to other calls, which may explain decreasing rates of ambulance dispatches. Potential solutions are likely to involve enhancing the flow through and discharge of patients from hospital, and a whole systems approach which considers the capacity of the local health and care infrastructure, including intermediate care and social care.

## Introduction

### Background

The challenges faced by the United Kingdom's National Health Service (NHS) before the COVID-19 pandemic have continued post-pandemic and are particularly apparent in urgent and emergency care (UEC). The target for the NHS is that 95% of emergency department (ED) attendances are seen and discharged, admitted, or transferred within four hours [1]. The NHS has not met the four-hour target in any year since 2013/14, and since July 2015 it has been missed every month [2]. UEC performance has declined markedly over the last five years; nationally: the percentage of Type 1 (ED) attendances admitted, transferred or discharged within four hours or less was 55.4% in November 2023, compared to 81.2% in November 2018 [3]. Being admitted from a crowded ED is associated with increased length of stay and may contribute to avoidable error, poor staff experience and increased mortality [4].

Urgent care involves any non-life-threatening illness or injury needing urgent attention which might be dealt with by phone consultation through the NHS 111 Clinical Assessment Service, pharmacy advice, out-of-hours general practitioner (GP) appointments, or at a minor injury clinic or walk-in centre. Emergency care involves life-threatening illnesses or accidents which require immediate treatment from an ED, often via an ambulance service (using the emergency healthcare call line, 999) [5]. We use the term "UEC" to cover all unscheduled urgent and emergency care-seeking at EDs.

COVID-19 may have affected demand for UEC by increasing unmet need by, for example, patients being unable or less willing to make GP appointments, under-detection of long-term conditions (for example, through reduced take up of NHS Health Checks), and delays to treatment [6–8]. In 2020, an estimated 400,000 planned procedures were not performed each month, which may have led to patients seeking UEC for health problems associated with delayed treatment [9]. COVID-19 also had a negative impact on NHS services through the reduction in staff wellbeing [10].

An interrupted time series analysis of the impact of COVID-19 on attendances and emergency and planned hospital admissions in Scotland found a significant drop in attendances (-41%) and emergency (-26%) and planned (-61%) admissions during the first few months of the pandemic [11]. A study of ED attendances at two large London hospitals found a decrease in attendances of 35% during the period after the first lockdown was imposed (12 March– 31 May 2020), reflecting broader trends seen across England [12]. Honeyford et al, in a retrospective study of changes in ED activity in a London hospital trust, found a reduction of 52% in daily ED attendance rates during lockdown (a four-week period in March/April 2020) [13]. COVID-19 and related lockdowns therefore resulted in short-term reductions in ED attendances. Less is known about UEC trends since the peak of the COVID-19 pandemic and associated lockdowns.

We have had the opportunity to explore UEC trends using publicly available data in the area served by Norfolk and Waveney (N&W) Integrated Care System, which is a partnership of organisations responsible for the provision of health care across the region. N&W is situated geographically in East Anglia in the UK and covers a population of almost 1.1 million [14]. Data covering the periods pre- and post-COVID-19 pandemic show substantial increases in ED waiting times (93% increase in those waiting more than four hours), alongside comparatively small increases in attendances (5%) [3]. Individual event patient level data was explored to examine in more details the impact of the COVID-19 pandemic and trends pre- and post-lockdown.

### Objectives

This research used individual event level patient data to explore the impact of COVID-19 on UEC in N&W. Specific objectives were to describe changes between and within pre-COVID, lockdown and post-lockdown periods in the numbers of:

- daily attendances at each of the three hospital EDs in N&W, and for the three EDs combined, overall and for subgroups of patients, and in the time patients spent in each ED before discharge, transfer or admission to hospital.

- ambulances called out (through NHS111 or 999) daily in N&W, and in the proportions of calls with subsequent delayed handovers to each ED.

### Methods

We used individual event level patient data to compare average ED attendances, waiting times, 999 calls and 111 calls in N&W between three time periods: pre-COVID, lockdown, and post-lockdown, along with continuous trends over time within each period. In this report 'post-lockdown' is defined as the period following the end of the stay-at-home rule of the final (third) lockdown in England, from 30 March 2021 to 31 March 2023. 'Pre-COVID' is used to define a period from 1 April 2018 to the start of the first lockdown on 25 March 2020. The lockdown period was from 26 March 2020 to 29 March 2021, and included three periods of lockdown:

- Lockdown 1: 26 March 2020 to 4 July 2020 inclusive

- Lockdown 2: 5 November 2020 to 2 December 2020 inclusive

- Lockdown 3: 6 January 2021 to 29 March 2021 inclusive

We carried out an interrupted time series analysis describing changes in UEC activity from April 2018 to March 2023. The time series analysis was carried out with health care activity data provided by the municipal health department for N&W Integrated Care System, the integrated care board (ICB). They collated data provided by NHS organisations including hospital trusts, the East of England Ambulance Service and NHS111. The raw data were at the level of individual health service visit, telephone call or ambulance call.

For the analyses we focused on Type 1 emergency care department data. Type 1 EDs are medical consultant-led 24-hour services with full resuscitation facilities. N&W has three Type 1 EDs, at the Norfolk and Norwich University Hospital (NNUH) in Norwich, the James Paget University Hospital (JPUH) in Great Yarmouth, and the Queen Elizabeth Hospital (QEH) in King's Lynn. We selected attendances where the patients were either registered with 105 GPs in N&W or had a home address in N&W (defined as 611 Lower layer Super Output Areas,

LSOA) and visited the main 24hr ED at NNUH, JPUH or QEH from 1 April 2018 to 31 March 2023. Activity for patients who were neither registered with a GP practice nor had an address located within N&W were excluded because the data included 'out of area' residents pre-COVID but not during the lockdown and post-COVID periods due to changes in reporting protocols. This particularly affects hospitals near county borders, for example QEH in King's Lynn, which sees more 'out of area' patients. Similarly, activity at 'out of area' hospitals by patients residing in N&W was excluded because our focus was on UEC in N&W.

At the NNUH, Type 1 attendances include the Children's Emergency Department and Older People's Emergency Department, but do not include 'Assessment Units'. This report excludes analysis of attendances to hospital same day emergency care centres, and general (family doctor) practices.

## Statistical methods

To describe changes in UEC activity over time we analysed the following variables, which were aggregated at day level using the individual event level patient data.

1. ED attendances: total number of attendances per day, arrival mode, diagnosis, referral source, and mean time spent in ED. Separate analyses were carried out for each ED in the study area, and for the aggregated N&W area data.

2. Ambulance dispatches: number and type of request for ambulance dispatch and counts of and proportions of ambulance to ED handover times greater than 60 minutes. Handover times were analysed separately for each ED, and for the aggregated area data.

3. We first compared mean values of each variable between pre-COVID, lockdown and post-lockdown periods using ANOVA tests. We then applied interrupted time series linear regression models to estimate changes in each variable. In all models we adjusted for day of the week and month of the year to account for seasonal and within-week variation in activity.

   For each outcome variable we used three sets of models:

- Model 1 estimates the change in mean values of each variable from pre-COVID to lockdown and post-lockdown periods.

- In Model 2 we additionally estimated continuous time trends during the pre-COVID, lockdown and post-lockdown periods. We also estimated the interaction between the time trend and the indicator of pre-COVID, lockdown and post-lockdown periods, to estimate how the continuous time trend changed after the lockdown period.

- In Model 3, we additionally included indicators of the three lockdown periods, to estimate the temporary effect of each lockdown on UEC activity rates.

With these data it was not possible for us to investigate individual characteristics associated with receipt of care, because the outcome variables for all of our analyses are aggregated, and because the individual level data are only available for individuals who received care, and not all individuals in the N&W population. To investigate potential differences in time trends between age groups, however, we graphed counts of monthly visits to EDs, stratified by age group.

Full details of methods are provided in S1 File.

We used a 5% significance level with 95% confidence intervals.

All time series linear regression models were produced using R statistical software version 4.1.2 (2021-11-01) [15]. Where percentage changes are reported these have been calculated

using the unadjusted pre-COVID mean, and the lockdown and post-lockdown change (adjusted for day of week and month of year) estimated with Model 1.

### Research governance

Research ethics approval was granted by UEA Faculty of Medicine and Health Sciences Research Ethics Subcommittee (ETH2223-1729, 20/03/2023). Consent from participants was not obtained as data were analysed pseudonymously. Data Protection Impact Assessments were completed in accordance with data governance procedures at Norfolk County Council (NCC) for the use of the pseudonymised patient record data. The data were sublicensed data from N&W ICB and named UEA researchers delivering the project were provided with NCC laptops to analyse data in the secure NCC data environment. The UEA researchers were accountable to and supervised by the NCC Insight and Analytics service. No individual level personal data were removed or shared outside of the NCC secure data environment.

## Results

### Number and characteristics of ED attendances

Between 1 April 2018 and 31 March 2023 the 519,152 individuals in the data selected for analysis made 1,193,359 total ED attendances. Of these attendances, 48.4% were to NNUH, 29.2% to JPUH and 22.3% to QEH. 48.1% of the study population were male and 51.9% were female (54 attendances had unspecified gender), and 83.6% were White British ethnicity (using ethnicity categories from the 2001 census). We excluded 351,275 ED attendances by individuals who were not resident or registered with a general practice in N&W, or who attended an ED outside N&W.

During this period, 98.5% of all ED attendances were for the first time for an incident and the remainder were for planned or unplanned subsequent attendances at the same department for the same incident as the first attendance. The most common reasons for ED attendance were injury (29.5%), respiratory disease (8.6%), circulatory problems (8.1%) and digestive system problems (6.6%). Arrival was by ambulance for 34.1% of ED attendances. including helicopter/air ambulance. Around half (48.4%) of attendances were self-referred to EDs, 27.0% were referred by ambulance service, 8.8% were referred by primary care health team and 7.8% were referred by NHS 111 service.

We analysed only data for those ambulances arriving at the three N&W trusts (NNUH, JPUH and QEH) from 2018/19 to 2022/23. The number of ambulances arriving at NNUH, JPUH and QEH were 244,359, 117,542 and 88,812, respectively, during this period. Of all ambulance calls 78.9% (n = 355,770) were initiated through 999 telephone calls and 21.1% (94,943) were initiated through NHS 111 calls.

Results of time series regression Models 1 and 2 are reported in the manuscript and results of Model 3 are confined to the Supporting Information. Regression coefficients were all adjusted by day of the week and month of the year. Outcomes are reported as percentage differences compared to the pre-COVID means shown in Table 1 (model 1 intercepts are inappropriate denominators for calculating percentage differences because they apply only to reference days and months).

### Numbers of daily ED attendances

Mean daily total ED attendances in N&W decreased by 10% from the pre-COVID period to the lockdown period, then increased in the post-lockdown period to be 7% more than pre-COVID (Tables 2 and S2, Figs 1 and S1). These overall changes were primarily due to changes

**Table 1. Mean values of urgent and emergency care indicators in Norfolk and Waveney before, during and after COVID lockdown periods.**

| Variable | Pre-COVID | Lockdown | Post-lockdown | p-value* |
|---|---|---|---|---|
| | Mean number (SD) per day | | | |
| All visits | 650 (68) | 582 (91) | 693 (65) | <0.001 |
| Arrive by ambulance | 234 (19) | 235 (22) | 205 (28) | <0.001 |
| Arrive by other | 415 (60) | 348 (76) | 488 (63) | <0.001 |
| Referred by primary care | 53 (28) | 49. (27) | 65 (33) | <0.001 |
| Referred by NHS 111 | 52 (17) | 58 (21) | 46 (11) | <0.001 |
| All ambulance callouts | 265 (19) | 255 (26) | 225 (33) | <0.001 |
| Ambulance callouts through NHS 111 | 60 (18) | 61 (17) | 40 (15) | <0.001 |
| Ambulance callouts 999 | 205 (19) | 193 (26) | 185 (27) | <0.001 |
| | Mean (SD) | | | |
| Minutes at ED | 220 (30) | 220 (31) | 320 (55) | <0.001 |
| Minutes at ED (ambulance arrivals) | 300 (53) | 304 (55) | 518 (129) | <0.001 |
| Minutes at ED (other arrivals) | 175 (22) | 162 (19) | 240 (39) | <0.001 |
| Ambulance handover > 60 minutes (%) | 7.5 (6.4) | 3.6 (4.4) | 24 (15) | <0.001 |

*ANOVA. ED emergency department. SD standard deviation.

in ED visits that did not arrive by ambulance, which were 16% lower during lockdown and 17% higher post-lockdown, compared to pre-COVID. Ambulance arrivals did not change from pre-COVID to lockdown, and were 13% lower post-lockdown than pre-COVID (Table 2). ED visits for injuries decreased by 16% from pre-COVID to lockdown, then increased to be 6% higher post-lockdown than pre-COVID. ED visits for circulatory diseases changed very little (3%) from pre-COVID to lockdown, and increased to be 15% higher post-lockdown than pre-COVID. ED visits due to referrals from primary care decreased by 8% from pre-COVID to lockdown, to be 23% higher post-lockdown than pre-COVID. In contrast, ED visits due to referrals from NHS111 increased by 12% from pre-COVID to lockdown, then decreased be 8% higher post-lockdown than pre-COVID.

Estimates of changes in levels and in continuous time trends (slopes) for numbers of daily ED attendances are shown in Tables 3 and S3, Figs 1 and S1. Total daily ED visits in N&W increased continually by 64 daily visits per year pre-COVID, then dropped to a much lower

**Table 2. Changes in mean number of daily emergency department attendances in Norfolk and Waveney from pre-COVID period to COVID lockdown period, and from pre-COVID period to post-lockdown period.**

| Outcome | All visits | | | Ambulance arrivals | | | Non-ambulance arrivals | | |
|---|---|---|---|---|---|---|---|---|---|
| Covariates | Coefficient | 95% CI | p-value | Coefficient | 95% CI | p-value | Coefficient | 95% CI | p-value |
| Intercept (pre-COVID mean) | 585 | (573, 598) | <0.001 | 231 | (226.72, 236.08) | <0.001 | 354 | (343, 365) | <0.001 |
| Change pre- COVID to lockdown | -65 | (-75, -60) | <0.001 | 0.13 | (-2.76, 3.01) | 0.93 | -68 | (-75, -61) | <0.001 |
| Change pre—COVID to post-lockdown | 43 | (37,49) | <0.001 | -30 | (-32, -27) | <0.001 | 72 | (67, 78) | <0.001 |
| | Injury | | | Circulatory disease | | | Referred by primary healthcare team | | |
| Intercept (pre-COVID mean) | 173 | (168, 179) | <0.001 | 48 | (46, 50) | <0.001 | 62 | (59, 65) | <0.001 |
| Change pre- COVID to lockdown | -29 | (-32, -26) | <0.001 | 1.2 | (0.1, 2.3) | 0.03 | -4.0 | (-6.0, -2.1) | <0.001 |
| Change pre—COVID to post-lockdown | 10 | (8, 12) | <0.001 | 7.4 | (6.5, 8.3) | <0.001 | 12 | (11, 14) | <0.001 |
| | Referred by NHS111 | | | | | | | | |
| Intercept (pre-COVID mean) | 40 | (38, 43) | <0.001 | | | | | | |
| Change pre- COVID to lockdown | 6.1 | (4.7, 7.5) | <0.001 | | | | | | |
| Change pre—COVID to post-lockdown | -6.0 | (-7.1, -4.9) | <0.001 | | | | | | |

Linear regression models, adjusted for day of week and month of year (model 1). CI confidence interval.

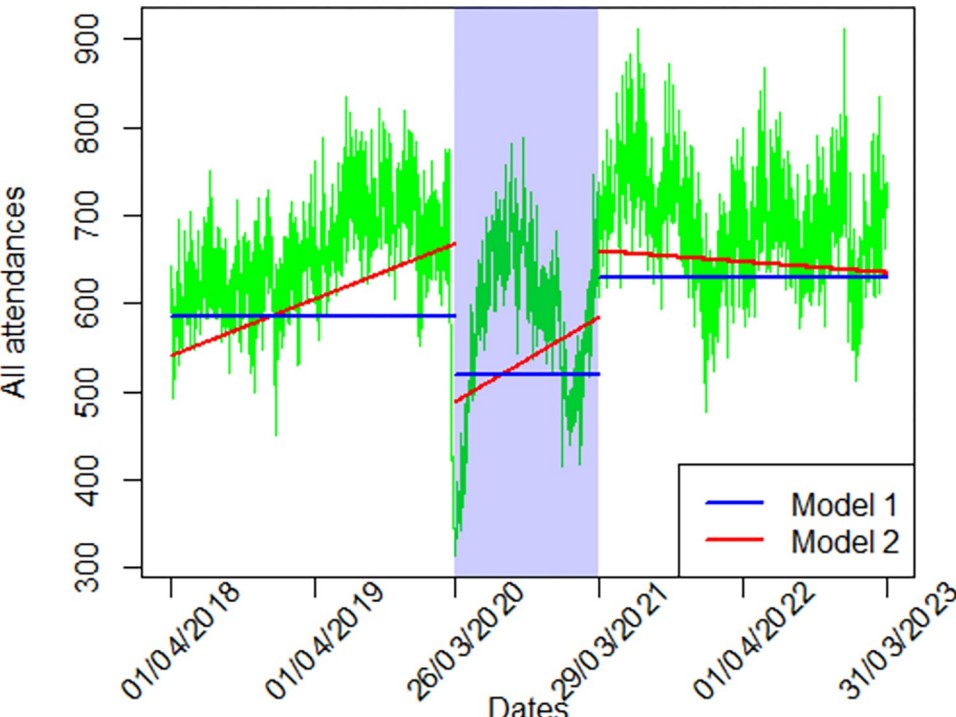

**Fig 1. Total daily visits in Norfolk and Waveney: Observations and statistically modelled changes.** Green lines connect daily observations, blue lines represent Model 1 means, and red lines represent Model 2 levels and slopes. The shaded area represents the lockdown period, the area to the left represents the pre-COVID period and the area to the right represents the post-lockdown period.

level during lockdown, during which they then increased by 95 daily visits per year. They then increased post-lockdown to a similar level to pre-COVID but decreased gradually during post-lockdown period (Fig 1 and Table 2). These time trends were similar in the three EDs (S2 Fig and S2 Table). ED visits arriving by ambulance increased from lockdown to post-lockdown periods, then decreased by 38 daily visits per year during the post-lockdown period. In contrast ED visits that did not arrive by ambulance continued to increase gradually by 26 daily visits per year post-lockdown. Other categories of ED visits either increased gradually post-lockdown (injuries and primary care referrals) or changed very little (circulatory diseases) or not at all (NHS111 referrals) within the post-lockdown period.

These results were similar for the three EDs, except for the following. ED visits referred from primary care increased during lockdown and post-lockdown in NNUH, but decreased in JPUH and QEH (S2 Table). Post-lockdown trends in non-ambulance arrivals increased more steeply in NNUH than in JPUH or QEH (S2 Table).

Time trends in total ED attendances were similar in all age groups (S2 Fig).

## Waiting times in ED

Waiting times in EDs were defined as the daily mean minutes from arrival at ED to discharge or admission to hospital. Changes in mean values are reported in Tables 4 and 5, S4 and S5 Tables. Mean waiting times for all visits did not change from pre-COVID to lockdown periods, but increased by 45% (100 minutes) from pre-COVID to post-lockdown. Mean waiting times for visits that followed ambulance arrivals increased by 72% (217 minutes), while visits that arrived by other means increased by 37% (65 minutes) (Table 4). Estimates of changes in levels

**Table 3. Change in level and in slope\* in mean number of daily emergency department attendances in Norfolk and Waveney, between pre-COVID, lockdown and post-lock periods.**

| Covariates | Coefficient | 95% CI | p-value | Coefficient | 95% CI | p-value | Coefficient | 95% CI | p-value |
|---|---|---|---|---|---|---|---|---|---|
| Outcome | All visits | | | Ambulance arrival | | | Non-ambulance arrival | | |
| Slope pre-COVID | 64 | (56, 712) | <0.001 | 3.1 | (0.60, 5.6) | 0.02 | 61 | (54, 68) | <0.001 |
| Change in level pre-COVID to lockdown | -242 | (-300, -186) | <0.001 | -50 | (-68.6, -326) | <0.001 | -192 | (-239, -144) | <0.001 |
| Change in slope pre-COVID to lockdown | 31 | (8.8, 54) | <0.001 | 18 | (12, 26) | <0.001 | 13 | (-6.2 32) | 0.18 |
| Slope during lockdown | 95 | (73, 118.) | <0.001 | 21 | (14.19, 28.58) | <0.001 | 74 | (55, 93) | <0.001 |
| Change in level pre-COVID to post-lockdown | 156 | (124, 485) | <0.001 | 125 | (115, 135) | <0.01 | 30 | (3.6, 575) | 0.026 |
| Change in slope pre-COVID to post-lockdown | -76 | (-87, 66) | <0.001 | -41 | (-44., 38) | <0.001 | -35 | (-44, -27) | <0.001 |
| Slope post-lockdown | -12 | (-2., -4.5) | 0.002 | -38 | (-40, -36) | <0.001 | 26 | (19, 33 | <0.001 |
| Outcome | Injury | | | Circulatory disease | | | Referred by primary healthcare team | | |
| Slope pre-COVID | 12 | (9.1, 16) | <0.001 | 3.3 | (2.1, 4.4) | <0.001 | 11 | (9.2, 13) | <0.001 |
| Change in level pre-COVID to lockdown | -147 | (-171, -123) | <0.001 | -13 | (-21, -4.7) | 0.002 | 4.7 | (-9.9, 19) | 0.53 |
| Change in slope pre-COVID to lockdown | 40 | (30, 50) | <0.001 | 3.8 | (0.4, 7.2) | 0.030 | -10 | (-16, -4.4) | <0.001 |
| Slope during lockdown | -4.9 | (-9.3, -0.5) | 0.029 | 7.0 | (3.6, 10) | <0.001 | 1.0 | (-4.9, 6.9) | 0.74 |
| Change in level pre-COVID to post-lockdown | -8.0 | (-22, 5.5) | 0.25 | 17 | (13) | <0.001 | 2.2 | (-6.0, 10) | 0.60 |
| Change in slope pre-COVID to post-lockdown | 52 | (43, 62) | <0.001 | -4.9 | (-6.4, -3.3) | <0.001 | -6.0 | (-8.6, -3.3) | <0.001 |
| Slope post-lockdown | 7.5 | (4.2, 10) | <0.001 | -1.6 | (-2.8, -0.5) | 0.006 | 5.3 | (3.3, 7.3) | <0.001 |
| Outcome | Referred by NHS111 | | | | | | | | |
| Slope pre-COVID | 7.3 | (5.8, 8.7) | <0.001 | | | | | | |
| Change in level pre-COVID to lockdown | 25 | (14, 35) | <0.001 | | | | | | |
| Change in slope pre-COVID to lockdown | -12 | (-17, -7.5) | <0.001 | | | | | | |
| Slope during lockdown | -4.5 | (-8.8, -0.3) | 0.04 | | | | | | |
| Change in level pre-COVID to post-lockdown | 1.0 | (-5.0, 6.9) | 0.75 | | | | | | |
| Change in slope pre-COVID to post-lockdown | -7.2 | (-9.13–5.3) | <0.001 | | | | | | |
| Slope post-lockdown | 0.1 | (-1.4, 1.5) | 0.93 | | | | | | |

Linear regression models, adjusted for day of week and month of year (model 2). \*Continuous increase or decrease per year. CI confidence interval.

and in continuous time trends showed small decreases in level from pre-COVID to lockdown, large decreases in level from pre-COVID to post-lockdown, and continuously increasing slopes during each of the three periods (Tables 5 and S5). The most steeply increasing trends were during the post-lockdown period (increasing by 60 minutes per year for all visits), and particularly in ambulance arrivals (increasing by 161 minutes per year). NNUH had the greatest increase in mean waiting times post-lockdown, especially for ambulance arrivals (S4 Table). However, the continuous upward trend in waiting times post-lockdown was steepest for QEH, followed by JPUH (S5 Table).

**Table 4. Changes in waiting times (mean number of minutes from arrival at emergency department until discharge or admission) in Norfolk and Waveney, from pre-COVID period to COVID lockdown period, and from pre-COVID period to post-lockdown period.**

| Outcome | All visits | | | Ambulance arrival | | | Non-ambulance arrival | | |
|---|---|---|---|---|---|---|---|---|---|
| Covariates | Coefficient | 95% CI | p-value | Coefficient | 95% CI | p-value | Coefficient | 95% CI | p-value |
| Intercept (pre-COVID mean) | 191 | (184, 199) | <0.001 | 234 | (217, 251) | <0.001 | 156 | (150, 161) | <0.001 |
| Pre-COVID to lockdown | -0.4 | (-5.1, 4.3) | 0.87 | 2.7 | (-7.7, 13.1) | 0.61 | -13 | (-17, -9.7) | <0.001 |
| Pre-COVID to post-lockdown | 100 | (96, 103) | <0.001 | 217 | (208, 225) | <0.001 | 65 | (62, 68) | <0.001 |

Linear regression models adjusted for day of week and month of year (model 1). CI confidence interval.

**Table 5. Change in level and in slope\* in minutes waiting at emergency departments in Norfolk and Waveney, between pre-COVID, lockdown and post-lock periods.**

| Outcome | All visits | | | Ambulance arrival | | | Non-ambulance arrival | | |
|---|---|---|---|---|---|---|---|---|---|
| Covariates | Coefficient | 95% CI | p-value | Coefficient | 95% CI | p-value | Coefficient | 95% CI | p-value |
| Slope pre-COVID | 26 | (22, 302) | <0.001 | 48 | (39, 56) | <0.001 | 21 | (18, 25) | <0.001 |
| Change in level pre-COVID to lockdown | -45 | (-75, -1)6 | 0.003 | -94 | (-153, -35) | 0.002 | -26 | (-49, -2.4) | 0.03 |
| Change in slope pre-COVID to lockdown | 2.5 | (-9.4, 14.) | 0.68 | 10 | (-14, 34) | 0.40 | -7.7 | (-17, 1.8) | 0.11 |
| Slope during lockdown | 29 | (17, 40) | <0.001 | 58 | (34, 82 | <0.001 | 14 | (4.2 23) | 0.0048 |
| Change in level pre-COVID to post-lockdown | -115 | (-132, -9.2) | <0.001 | -384 | (-417, -351) | <0.001 | -55 | (-69, -44) | <0.001 |
| Change in slope pre-COVID to post-lockdown | 34 | (29, 39.44 | <0.001 | 114 | (104, 125) | <0.001 | 14 | (9.9, 18) | <0.001 |
| Slope post-lockdown | 60 | (56, 64) | <0.001 | 161 | (154, 170) | <0.001 | 35 | (32, 39) | <0.001 |

Linear regression models adjusted for day of week and month of year (model 2). \*Continuous increase or decrease per year. CI confidence interval.

## Ambulance callouts

Mean daily numbers of ambulance callouts in N&W decreased by 4% from pre-COVID to lockdown, and decreased by 15% from pre-COVID to post-lockdown (Table 6). Estimates of changes in levels and in continuous time trends showed no change in slope pre-COVID, a lower level with increasing slope during lockdown, and a higher level with decreasing slope post-lockdown (Table 7). During the post-lockdown period mean daily ambulance callouts were decreasing steadily by 44 callouts per year (Table 7). These changes were similar for ambulance callouts from NHS111 and from 999 telephone calls. Ambulance callout data are not reported separately for each ED because the ambulance service covers all of N&W.

## Ambulance handover delays

Delays in ambulance handovers to EDs of more than 60 minutes were uncommon pre-COVID (7.5%, Table 1). Changes in the frequency of such delays are reported in Tables 8 and 9, S6 and S7 Tables. Delays decreased by 4% from pre-COVID to lockdown periods, but increased by 17% from pre-COVID to post-lockdown (Table 8). During the post-lockdown period the proportion of ambulance handovers taking over 60 minutes increased steadily by 20% per year, in contrast to the pre-COVID and lockdown periods during which there was minimal continuous change (Table 9).

## Discussion

ED attendances increased slightly in the post-lockdown period compared to the pre-COVID period. These increases were due to patients who did not arrive by ambulance. Waiting times at ED increased substantially from pre-COVID to post-lockdown and continued to increase

**Table 6. Changes in mean number of ambulance callouts in Norfolk and Waveney from pre-COVID period to COVID lockdown period, and from pre-COVID period to post-lockdown period.**

| Outcome | All ambulance calls | | | Called through NHS 111 | | | Called through 999 | | |
|---|---|---|---|---|---|---|---|---|---|
| Covariates | Coefficient | 95% CI | p-value | Coefficient | 95% CI | p-value | Coefficient | 95% CI | p-value |
| Intercept (pre-COVID mean) | 265 | (260, 702) | <0.001 | 52 | (50, 55) | <0.001 | 212 | (208, 217) | <0.001 |
| Pre-Covid to lockdown | -10 | (-13, -6.5) | <0.001 | 1.7 | (0.3, 3.2) | 0.02 | -12 | (-14, -8.9) | <0.001 |
| Pre-Covid to post-lockdown | -39 | (-42, -37) | <0.001 | -20 | (-21, -19) | <0.001 | -20 | (-22, -17) | <0.001 |

Linear regression models, adjusted for day of week and month of year (model 1). CI confidence interval.

**Table 7. Changes in level and in slope\* in mean number of ambulance callouts in Norfolk and Waveney, between pre-COVID, lockdown and post-lockdown periods.**

| Outcome | All ambulance calls | | | Called through NHS 111 | | | Called through 999 | | |
|---|---|---|---|---|---|---|---|---|---|
| Covariates | Coefficient | 95% CI | p-value | Coefficient | 95% CI | p-value | Coefficient | 95% CI | p-value |
| Slope pre-COVID | -1.0 | (-3.8, 1.8) | 0.48 | 0.5 | (-1.0, 2.0) | 0.54 | -1.5 | (-3.8, 0.9) | 0.21 |
| Change in level pre-COVID to lockdown | -89 | (-109, -69) | <0.001 | -30 | (-41, -19) | <0.001 | -59 | (-76, -42) | <0.001 |
| Change in slope pre-COVID to lockdown | 32 | (24, 40) | <0.001 | 12 | (8.0, 17) | <0.001 | 20 | (13, 27) | <0.001 |
| Slope during lockdown | 31 | (23, 39) | <0.001 | 13 | (8.4, 17) | <0.001 | -18 | (12, 25) | <0.001 |
| Change in level pre-COVID to post-lockdown | 137 | (126, 148) | <0.001 | 21 | (15, 27) | <0.001 | 116 | (107, 126) | <0.001 |
| Change in slope pre-COVID to post-lockdown | -43 | (-47, -40) | <0.001 | -11 | (-13, -8.6) | <0.002 | -33 | (-36, -30) | <0.001 |
| Slope post-lockdown | -44 | (-47, -42) | <0.001 | -10 | (-12, -8.6) | <0.001 | -34 | (-37, -32) | <0.001 |

Linear regression models, adjusted for day of week and month of year (model 2). \*Continuous increase or decrease per year. CI confidence interval.

steadily during the post-lockdown period (except in NNUH where they decreased during the most recent 2022–23 year). Patients were also more likely to have delayed handovers of more than 60 minutes from ambulance to hospital care.

The greatly increased waiting times at ED and for ambulance handovers cannot be explained solely by the increase in attendances. Post lockdown, ED patients faced long waits to be admitted, discharged or transferred. This results in ambulances being delayed handing over patients and attending to other calls, which may explain decreasing rates of ambulance dispatches; this has been a national problem since COVID-19 [16,17].

It has been claimed that the main reason that EDs are unable to promptly treat, discharge or admit patients is the lack of available hospital beds [18,19]. Analysis of patient pathways and flow within EDs and hospitals was, however, beyond the scope of this study. Publicly available data show that the number of doctors in N&W EDs has increased by between 40% and 262% since 2018, and the number of nurses by 24–62%, indicating that long waiting times cannot be remedied by solely increasing ED staff capacity [20,21]. More prompt discharge of some cohorts of patients may be enabled by increasing social care capacity locally, which data suggest may have decreased in recent years [22]. Other potential solutions are likely to involve the broader health and social care system (including mental health services and intermediate care services) in the short and long-term.

The results of the present study suggest that COVID and COVID lockdowns do not appear to have had major lasting effects on attendances at UEC in N&W during the post-lockdown period. The UEC system largely recovered from the shock of COVID and related lockdowns, with most activity returning to similar levels post-lockdown as pre-COVID. Those indicators that became progressively worse post-lockdown could be due to tightening bottlenecks in

**Table 8. Changes in mean percentage of ambulance callouts with delayed handover to emergency departments of more than 60 minutes in Norfolk and Waveney, from pre-COVID period to COVID lockdown period, and from pre-COVID period to post-lockdown period.**

| Outcome | % handovers delayed >60 minutes | | |
|---|---|---|---|
| Covariates | Coefficient | 95% CI | p-value |
| Intercept (pre-COVID mean) | -0.5 | (-2.4, 1.5) | 0.63 |
| Pre-COVID to lockdown | -4.0 | (-5.2, -2.8) | <0.001 |
| Pre-COVID to post-lockdown | 17 | (16, 18) | <0.001 |

Linear regression models adjusted for day of week and month of year (model 1). CI confidence interval.

**Table 9. Changes in level and change in slope\* in percentage of ambulance callouts with delayed handover to emergency departments of more than 60 minutes in Norfolk and Waveney, between pre-COVID, lockdown and post-lock periods.**

| Outcome | % handovers delayed >60 minutes | | |
|---|---|---|---|
| Covariates | Coefficient | 95% CI | p-value |
| Slope pre-COVID | 0.8 | (-0.1, 164) | 0.09 |
| Change in level pre-COVID to lockdown | 0.4 | (-5.9, 6.7) | 0.89 |
| Change in slope pre-COVID to lockdown | -2.2 | (-4.8, 0.3) | 0.09 |
| Slope during lockdown | -1 | (-4.0, 1.1) | 0.27 |
| Change in level pre-COVID to post-lockdown | -63 | (-67, -60) | <0.001 |
| Change in slope pre-COVID to post-lockdown | 19 | (18, 21) | <0.001 |
| Slope post-lockdown | 20 | (19, 21) | <0.001 |

Linear regression models adjusted for day of week and month of year (model 2). *Continuous increase or decrease per year. CI confidence interval.

patient flow during ED care, admission to and discharge from hospital. As we did not have data on inpatient flow through hospitals to discharge and to social care, we were unable to investigate these processes.

## Strengths and limitations

Differences between results obtained using the three regression models should be considered. Model 1 results are most informative in showing the changes in average UEC from the pre-COVID to the lockdown periods, and how this activity generally returned to levels post-lockdown that were similar to pre-COVID levels; exceptions were ED waiting times, ambulance arrivals and callouts, and ambulance handover times which changed substantially. Model 2 results added to these findings by describing the continuous time trends in UEC within pre-COVID and post-lockdown periods. They showed gradually increasing demand for UEC pre-COVID, and steeply changing continuous time trends post-lockdown for ED waiting times, ambulance callouts and ambulance handover delays. Model 3 results provided additional information on the magnitude of temporary decreases in every UEC activity in every ED in each of the three lockdowns (S8 Table).

A strength of this study is the size, detail and variety of the UEC dataset that was used for the analyses. The range of the data provided allowed us to get a comprehensive picture of the impact of COVID-19 on UEC, looking not just at ED attendances but at waiting times and requests for ambulance dispatches. Data provided at the daily level allowed us to examine the specific impact of lockdown periods and changes to attendances/waiting times/dispatches during these periods.

The study has used routine data sources (Hospital Episode Statistics and the Emergency Care Data Set) which, as medical records, have limitations when used for research. A limitation is that due to differences and changes in data recording, results are not entirely comparable over the years, and we were unable to carry out similar analyses of Type 3 urgent and emergency care (attendances at walk-in centres, minor injuries units, and GP Front Door services). Data for attendance have only been collected by those with an address or registered general practice located in N&W, therefore excluding attendances from residents out of area. This has particularly affected QEH in Kings Lynn which lies on the border of West Norfolk, South Lincolnshire and East Cambridgeshire, serving a large proportion of out of area patients. Findings may not be generalisable beyond the specific area of study. However, other areas of the

United Kingdom are likely to have had similar experiences and may be able to draw lessons from this research. It would have been desirable to investigate individual level characteristics associated with changes in UEC activity, but that was not possible as these data excluded N&W residents who did not receive UEC. In a separate study we analysed cross-sectional data on associations between sociodemographic characteristics of every N&W resident, and their number of ED visits during one year post-lockdown [23]. This found that socioeconomic deprivation and chronic health conditions were major determinants of ED attendance. Future research would benefit from linkage of longitudinal data on every individual in a geographical area with longitudinal individual level data on UEC.

## Conclusion

This research confirms previous evidence that COVID-19 led to a reduction in emergency department attendances during lockdown periods, followed by an increased demand. It adds important new information about activity and trends pre-COVID, during and post-lockdown, sources of referral to EDs, and ambulance waiting times. Greatly increased waiting times and ambulance handover times at EDs are not wholly explained by the relatively small increases in numbers of attendances, suggesting that improvements are likely to require better flow through and discharge from hospital, to enable prompt admissions, rather than solely investing in increased emergency department capacity. Further research is needed to understand bottlenecks in UEC pathways, including the availability of beds, and the impact that delayed discharge and availability of social care provision may be having on capacity and flow.

## Supporting information

**S1 Table. Mean values of urgent and emergency care indicators in each hospital before, during and after COVID lockdown periods.** P<0.001 for all comparisons between the three periods with ANOVA test. ED emergency department. SD standard deviation.
(DOCX)

**S2 Table. Changes in mean number of daily emergency department attendances in each hospital from pre-COVID period to COVID lockdown period, and from pre-COVID period to post-lockdown period.** Linear regression models, adjusted for day of week and month of year (model 1). NNUH Norfolk and Norwich University Hospital. JPUH James Paget University Hospital. QEH Queen Elizabeth Hospital. CI confidence interval.
(DOCX)

**S3 Table. Change in level and change in slope (continuous increase or decrease per year) in mean number of daily emergency department attendances in each hospital, during pre-COVID, lockdown and post-lock periods.** Linear regression models, adjusted for day of week and month of year (model 2). NNUH Norfolk and Norwich University Hospital. JPUH James Paget University Hospital. QEH Queen Elizabeth Hospital. CI confidence interval. CI confidence interval.
(DOCX)

**S4 Table. Differences in waiting times (mean number of minutes from arrival at emergency department until discharge or admission) in each hospital, from pre-COVID period to COVID lockdown period, and from pre-COVID period to post-lockdown period.** Linear regression models adjusted for day of week and month of year (model 1). NNUH Norfolk and Norwich University Hospital. JPUH James Paget University Hospital. QEH Queen Elizabeth Hospital. CI confidence interval.
(DOCX)

**S5 Table. Change in level and change in slope (continuous increase or decrease per year) in minutes waiting at emergency departments in each hospital, during pre-COVID, lockdown and post-lock periods.** Linear regression models adjusted for day of week and month of year (model 2). NNUH Norfolk and Norwich University Hospital. JPUH James Paget University Hospital. QEH Queen Elizabeth Hospital. CI confidence interval.
(DOCX)

**S6 Table. Differences in mean percentage of ambulance callouts with delayed handover to emergency departments of more than 60 minutes in each hospital, from pre-COVID period to COVID lockdown period, and from pre-COVID period to post-lock down period.** Linear regression models adjusted for day of week and month of year (model 1). NNUH Norfolk and Norwich University Hospital. JPUH James Paget University Hospital. QEH Queen Elizabeth Hospital. CI confidence interval.
(DOCX)

**S7 Table. Change in level and change in slope (continuous increase or decrease per year) in percentage of ambulance callouts with delayed handover to emergency departments of more than 60 minutes in each hospital, during pre-COVID, lockdown and post-lock periods.** Linear regression models adjusted for day of week and month of year (model 2). NNUH Norfolk and Norwich University Hospital. JPUH James Paget University Hospital. QEH Queen Elizabeth Hospital. CI confidence interval.
(DOCX)

**S8 Table. Temporary changes in numbers of visits to each emergency department during first, second and third lockdowns.** Linear regression models adjusted for time trend and lockdown-trend interaction (model 3). NNUH Norfolk and Norwich University Hospital. JPUH James Paget University Hospital. QEH Queen Elizabeth Hospital. CI confidence interval.
(DOCX)

**S1 Fig. Total daily visits in each emergency department: Observations and statistically modelled changes.** Green lines connect daily observations, blue lines represent Model 1 means, and red lines represent Model 2 levels and slopes. The shaded area represents the lockdown period, the area to the left represents the pre-COVID period and the area to the right represents the post-lockdown period. NNUH Norfolk and Norwich University Hospital. JPUH James Paget University Hospital. QEH Queen Elizabeth Hospital. CI confidence interval.
(TIF)

**S2 Fig. Total monthly visits to each emergency department, and to all three emergency departments aggregated, for different age groups.** The shaded area represents the lockdown period, the area to the left represents the pre-COVID period and the area to the right represents the post-lockdown period. NNUH Norfolk and Norwich University Hospital. JPUH James Paget University Hospital. QEH Queen Elizabeth Hospital. CI confidence interval.
(TIF)

**S1 File.**
(DOCX)

## Author Contributions

**Conceptualization:** Max Bachmann, Charlotte Emily Louise Jones, Oby Otu Enwo, Alice M. Dalton, Nicholas Steel.

**Data curation:** Zillur Rahman Shabuz, Max Bachmann, Amanda Burke.

**Formal analysis:** Zillur Rahman Shabuz, Max Bachmann.

**Funding acquisition:** Max Bachmann, Nicholas Steel.

**Investigation:** Zillur Rahman Shabuz, Max Bachmann.

**Methodology:** Zillur Rahman Shabuz, Max Bachmann, Nicholas Steel.

**Project administration:** Amanda Burke, Nicholas Steel.

**Supervision:** Amanda Burke, Nicholas Steel.

**Validation:** Zillur Rahman Shabuz.

**Visualization:** Zillur Rahman Shabuz.

**Writing – original draft:** Rachel Cullum, Amanda Burke, Charlotte Emily Louise Jones.

**Writing – review & editing:** Zillur Rahman Shabuz, Max Bachmann, Rachel Cullum, Amanda Burke, Charlotte Emily Louise Jones, Oby Otu Enwo, Alice M. Dalton, Julii Brainard, Michael Saunders, Nicholas Steel.

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
