## [Decision Letter · Decision Letter 0]

6 Feb 2024

PONE-D-23-42956Changes in urgent and emergency care activity associated with COVID-19 lockdowns in a sub-region in the East of England: interrupted times series analyses .PLOS ONE

Dear Dr. Cullum,

Thank you for submitting your manuscript to PLOS ONE. After careful consideration, we feel that it has merit but does not fully meet PLOS ONE’s publication criteria as it currently stands. Therefore, we invite you to submit a revised version of the manuscript that addresses the points raised during the review process.

Please address reviewers' comments==============================

We look forward to receiving your revised manuscript.

Kind regards,

Chunyu Liu, PhD

Academic Editor

PLOS ONE

2. For studies involving third-party data, we encourage authors to share any data specific to their analyses that they can legally distribute. PLOS recognizes, however, that authors may be using third-party data they do not have the rights to share. When third-party data cannot be publicly shared, authors must provide all information necessary for interested researchers to apply to gain access to the data. (https://journals.plos.org/plosone/s/data-availability#loc-acceptable-data-access-restrictions)

Reviewers' comments:

Reviewer's Responses to Questions

**Comments to the Author**

1. Is the manuscript technically sound, and do the data support the conclusions?

Reviewer #1: Yes

Reviewer #2: Partly

2. Has the statistical analysis been performed appropriately and rigorously? 

Reviewer #1: Yes

Reviewer #2: Yes

3. Have the authors made all data underlying the findings in their manuscript fully available?

Reviewer #1: Yes

Reviewer #2: No

4. Is the manuscript presented in an intelligible fashion and written in standard English?

Reviewer #1: Yes

Reviewer #2: Yes

5. Review Comments to the Author

Reviewer #1: The paper studies the changes in urgent and emergency care activity associated with the COVID-19 lockdowns using time series analysis. The introduction and method section are clear written. The supplementary files provided sufficient and clear details for the time series analysis. My comments are as follows:

1. The authors used data from three hospitals and performed the times-series separately. I suggest authors consider performing analysis after aggregating data together, since an aggregate analysis can identify overarching trends that are consistent across all hospitals

2. Following the last comment, the setting in the article appears to be a classical example in a textbook for mixed-effect models: a variety of collected outcomes were nested in hospitals. Since this approach could provide insights into both individual hospital effects and the overall effect across the system, I wonder if authors consider performing a mixed-effect model analysis for these data.

3. Though authors analyzed the data using three different models, when reporting results, majority of results reported are from the Model 1, and the connection and differentiation between those models are not discussed. I recommend authors discuss the results contrasting these three models.

4. By modeling the hospital data before and after the lock-down, would it be sufficient to conclude that such change is caused by, or associated with, the COVID lockdown? I wonder if authors would consider other confounders or factors that may impact the change in hospital data?

5. In addition to reporting the coefficient values in the text, I recommend authors use tables or plots to report the estimated values for a clearer understanding, especially for readers who may grasp visual information better than textual data.

Reviewer #2: The authors use interrupted time series analysis to compare use of urgent and emergency care (UEC) before the onset of the covid-19 pandemic with the use of UEC after the subsequent lockdowns in the U.K. Existing work had shown a reduction in use of hospitals during the early months of the pandemic, but had not yet considered the time after the peak of the covid-19 pandemic and the end of the U.K. lockdowns. They consider a large data set from a consortium of groups providing healthcare across East England and compare the daily attendances at hospitals, ambulances called out, and referrals made to emergency departments before the covid-19 pandemic (2018-2020) and after the subsequent lockdowns (2021-2023). The authors are addressing an interesting problem, but there could be more rigor in the methodology choices and justifications. I have some comments and suggestions for improvement below.

• It would be helpful to include the justification for exclusion of out of area patients and list the number of out of area patients excluded from each region.

• Similarly, it would be helpful to include a justification for analyzing the hospitals separately rather than using some sort of random effect analysis to borrow information across hospitals.

• Provide anova results for table 1 (in supplement if necessary). Provide demographic and descriptive information (gender, ethnicity, number of repeat visits, etc) separately by hospital if the analysis is also being provided separately by hospital.

• In models 1 and 2, why not also model the intermediate period between the lockdowns as a separate portion of the model, making it a three-level variable of pre, during, and post?

• Since the individual level data is available, there seems to be a missed opportunity to consider this question using individual level data, allowing for adjustment for demographic variables, repeated visits, etc. or comparison between groups. If not in the scope of this paper, some discussion of this possibility would be useful.

• In the discussion, reference number 20 does not appear to provide information about the number of doctors and nurses since 2018. Please correct or clarify the reference.

6. PLOS authors have the option to publish the peer review history of their article (what does this mean?). If published, this will include your full peer review and any attached files.

Reviewer #1: No

Reviewer #2: No

---

## [Author Response · Author response to Decision Letter 0]

22 Apr 2024

Response to Reviewers:

Reviewer #1: The paper studies the changes in urgent and emergency care activity associated with the COVID-19 lockdowns using time series analysis. The introduction and method section are clear written. The supplementary files provided sufficient and clear details for the time series analysis. My comments are as follows:

1. The authors used data from three hospitals and performed the times-series separately. I suggest authors consider performing analysis after aggregating data together, since an aggregate analysis can identify overarching trends that are consistent across all hospitals.

• Response: Thank you for this excellent suggestion. We have aggregated the data from all three hospitals and added the corresponding results to the manuscript. We have thoroughly revised the entire Results section, to avoid overloading readers with detail, as follows: In the manuscript text and tables we report results for analysis of the aggregated data, and have moved the tables and figures for each hospital to the supplementary appendix. In the text of Results we primarily comment on results pertaining to the aggregated data, and only refer to specific hospitals if their results differ substantially from the others. 

2. Following the last comment, the setting in the article appears to be a classical example in a textbook for mixed-effect models: a variety of collected outcomes were nested in hospitals. Since this approach could provide insights into both individual hospital effects and the overall effect across the system, I wonder if authors consider performing a mixed-effect model analysis for these data.

Response: Although it would be possible to analyse these data with mixed effect models (i.e. with outcomes for each day represented by three records – one for each hospital – and with hospitals modelled as random effects) we believe that in this study it is not appropriate to do so, for the following reasons:

• Firstly, we consider the analysis of the aggregated data (see above) to be the clearest way to represent the overall effect across the system.

• Secondly, there are only three hospitals, and they differ systematically from each other in the numbers of ED patients attending every day. That is, most of the outcomes are counts of numbers of events per day, and the three EDs are at different scales. Therefore the changes of interest – estimated in counts per day - vary systematically between the EDs, in proportion to their scale. The results of mixed models would be expressed as average changes in each ED which are not really meaningful. Although it would be possible in a mixed model to estimate different effects for each ED by adding covariates representing hospitals, together with hospital-time interaction terms, this would increase the complexity of these models (especially in models 2 and 3 which already have several interaction terms representing changes in slopes), make their results complex and difficult to interpret, not add anything to the original stratified analysis and the new additional analysis of aggregated data. 

3. Though authors analyzed the data using three different models, when reporting results, majority of results reported are from the Model 1, and the connection and differentiation between those models are not discussed. I recommend authors discuss the results contrasting these three models.

• Response: We have added Model 2 results for every UEC indicator, both in the Tables now included in the manuscript and in the accompanying text. We have simplified reporting of the Model 3 results, in supplementary table S8. 

• We have added a paragraph to Discussion: “Differences between results obtained using the three regression models should be considered. Model 1 results are most informative in showing the changes in average UEC from the pre-COVID to the lockdown periods, and how this activity generally returned to levels post-lockdown that were similar to pre-COVID levels; exceptions were ED waiting times, ambulance arrivals and callouts, and ambulance handover times which changed substantially. Model 2 results added to these findings by describing the continuous time trends in UEC within pre-COVID and post-lockdown periods. They showed gradually increasing demand for UEC pre-COVID, and steeply changing continuous time trends post-lockdown for ED waiting times, ambulance callouts and ambulance handover delays. Model 3 results provided additional information on the magnitude of temporary decreases in every UEC activity in every ED in each of the three lockdowns (S8 Table).”

4. By modeling the hospital data before and after the lock-down, would it be sufficient to conclude that such change is caused by, or associated with, the COVID lockdown? I wonder if authors would consider other confounders or factors that may impact the change in hospital data?

• Response: We have added to Discussion: “The results of the present study suggest that COVID and COVID lockdowns do not appear to have had major lasting effects on attendances at UEC in N&W during the post-lockdown period. The UEC system largely recovered from the shock of COVID and related lockdowns, with most activity returning to similar levels post-lockdown as pre-COVID. Those indicators that became progressively worse post-lockdown could be due to tightening bottlenecks in patient flow during ED care, admission to and discharge from hospital. As we did not have data on inpatient flow through hospitals to discharge and to social care, we were unable to investigate these processes.” 

5. In addition to reporting the coefficient values in the text, I recommend authors use tables or plots to report the estimated values for a clearer understanding, especially for readers who may grasp visual information better than textual data.

• Response: We have added tables of results (for aggregated data) to the manuscript. We have added Fig 1 to the Manuscript, to illustrate and help explain the results reported in the tables, and make them more intelligible by readers. To avoid excessive detail and duplication, however, we have not added more figures corresponding to each of the tables. 

Reviewer #2: The authors use interrupted time series analysis to compare use of urgent and emergency care (UEC) before the onset of the covid-19 pandemic with the use of UEC after the subsequent lockdowns in the U.K. Existing work had shown a reduction in use of hospitals during the early months of the pandemic, but had not yet considered the time after the peak of the covid-19 pandemic and the end of the U.K. lockdowns. They consider a large data set from a consortium of groups providing healthcare across East England and compare the daily attendances at hospitals, ambulances called out, and referrals made to emergency departments before the covid-19 pandemic (2018-2020) and after the subsequent lockdowns (2021-2023). The authors are addressing an interesting problem, but there could be more rigor in the methodology choices and justifications. I have some comments and suggestions for improvement below.

• It would be helpful to include the justification for exclusion of out of area patients and list the number of out of area patients excluded from each region.

• Response: We have added to Methods, “Activity for patients who were neither registered with a GP practice nor had an address located within N&W were excluded because the data included these out of area’ residents pre-COVID but excluded them during the lockdown and post-COVID periods due to changes in reporting protocols within the ICB.” And “Similarly, activity at ‘out of area’ hospitals by patients residing in N&W was excluded because our focus was on UEC in N&W.” 

• We have added the number of out of area residents to the first paragraph of Results, “We excluded 351,275 ED attendances by individuals who were not resident or registered with a general practice in Norfolk and Waveney, or who attended an ED outside Norfolk and Waveney.” Unfortunately we are unable to categorise excluded individuals by region. 

• Similarly, it would be helpful to include a justification for analyzing the hospitals separately rather than using some sort of random effect analysis to borrow information across hospitals.

• Please see our response to Reviewer #1, comment 2. 

• Provide anova results for table 1 (in supplement if necessary). 

• Response: We have added ANOVA p-values to Table 1. In the supplementary table S1 we have added to the legend: “P<0.001 for all comparisons between the three periods with ANOVA test”.

Provide demographic and descriptive information (gender, ethnicity, number of repeat visits, etc) separately by hospital if the analysis is also being provided separately by hospital.

• Unfortunately we no longer have access to these individual level data and so are unable to report these results. However, we have added graphs comparing time trends in ED attendances stratified by age group, in each hospital and in the area combined. 

• In models 1 and 2, why not also model the intermediate period between the lockdowns as a separate portion of the model, making it a three-level variable of pre, during, and post?

• Response: Thank you for this excellent suggestion. We have added these analyses and results. 

• Since the individual level data is available, there seems to be a missed opportunity to consider this question using individual level data, allowing for adjustment for demographic variables, repeated visits, etc. or comparison between groups. If not in the scope of this paper, some discussion of this possibility would be useful.

• Response: With these data it was not possible for us to investigate individual characteristics associated with receipt of care, because the outcome variables for all of our analyses are aggregated (e.g. number of visits to an ED each day), and because the individual level data are only available for individuals who received care, and not all individuals in the N&W population. 

• We have added to Methods, “Because outcome data were aggregated at ED or area level, it was not possible to adjust statistically for the characteristics of each individual patient. To investigate potential differences between age groups, we graphed counts of monthly visits to each ED, and to all EDs aggregated, stratified by age group.” We also added to Results: “Time trends in ED attendances were similar in all age groups (S2 Fig.)” and added S2 Fig. to the Supplementary Appendix. 

• We have added to Discussion, Strengths and limitations: “It would have been desirable to investigate individual level characteristics associated with changes in UEC activity, but that was not possible as these data excluded N&W residents who did not receive UEC. In a separate study, currently under review, we analysed cross-sectional data on associations between sociodemographic characteristics of every N&W resident, and their number of ED visits during one year post-lockdown. This found that socioeconomic deprivation and chronic health conditions were major determinants of ED attendance. Future research would benefit from linkage of longitudinal data on every individual in a geographical area with longitudinal individual level data on UEC.”

• In the discussion, reference number 20 does not appear to provide information about the number of doctors and nurses since 2018. Please correct or clarify the reference.

The reference has been corrected and replaced with the two correct references which are:

• 20. NHS Digital. Hospital Accident & Emergency Activity, 2018-19 . Tables 11a and 11c. Available from: https://files.digital.nhs.uk/07/14500F/AE1819_Provider_Level_Analysis.xlsx

• 21. NHS Digital. Hospital Accident & Emergency Activity, 2022-23. Tables 1a and 1c. Available from: https://files.digital.nhs.uk/02/4EF4B3/AE2223%20acci-emer-workforce-data.xlsx

---

## [Decision Letter · Decision Letter 1]

12 Sep 2024

PONE-D-23-42956R1Changes in urgent and emergency care activity associated with COVID-19 lockdowns in a sub-region in the East of England: interrupted times series analyses .PLOS ONE

Dear Dr. Bachmann,

Thank you for submitting your manuscript to PLOS ONE. After careful consideration, we feel that it has merit but does not fully meet PLOS ONE’s publication criteria as it currently stands. Therefore, we invite you to submit a revised version of the manuscript that addresses the points raised during the review process.

We look forward to receiving your revised manuscript.

Kind regards,

Niklas Bobrovitz

Academic Editor

PLOS ONE

Journal Requirements:

Additional Editor Comments:

Please address the following comments:

Abstract: Add 95% confidence intervals to the estimates

Tables/Figures: Add footnotes describing the abbreviations used in the data displays (e.g., hospital name)

Page 22, line 32: Typo - eliminate word “around”

Page 22, line 37: Type – increasing or decreasing?

Please proof read the entire mansucript again. There may have been some minor errors when accepting track changes from the first revision.

Reviewers' comments:

Reviewer's Responses to Questions

**Comments to the Author**

1. If the authors have adequately addressed your comments raised in a previous round of review and you feel that this manuscript is now acceptable for publication, you may indicate that here to bypass the “Comments to the Author” section, enter your conflict of interest statement in the “Confidential to Editor” section, and submit your "Accept" recommendation.

Reviewer #1: All comments have been addressed

Reviewer #2: All comments have been addressed

2. Is the manuscript technically sound, and do the data support the conclusions?

Reviewer #1: Yes

Reviewer #2: Yes

3. Has the statistical analysis been performed appropriately and rigorously? 

Reviewer #1: Yes

Reviewer #2: Yes

4. Have the authors made all data underlying the findings in their manuscript fully available?

Reviewer #1: Yes

Reviewer #2: No

5. Is the manuscript presented in an intelligible fashion and written in standard English?

Reviewer #1: Yes

Reviewer #2: Yes

6. Review Comments to the Author

Reviewer #1: All my comments have been well-addressed. In particular, authors redid the analyses using the aggregated data and reported the results accordingly. Great work!

Reviewer #2: The authors have provided a thorough response to the reviewer comments, and my comments have been addressed.

7. PLOS authors have the option to publish the peer review history of their article (what does this mean?). If published, this will include your full peer review and any attached files.

Reviewer #1: No

Reviewer #2: No

---

## [Author Response · Author response to Decision Letter 1]

23 Sep 2024

We have responded as follows to each of the Editor’s comments. The reviewer’s comments were all favourable and did not require any additional changes. (One reviewer noted that we had not submitted the study data; this was because, as stated in the Data Availability Statement, these data are confidential medical records which the authors are not authorised to share.)

Journal Requirements:

Response. We have added reference 23, and edited the relevant sentence in Discussion. In the previous submission we had written that the article was under review, but it has been published since then. 

Additional Editor Comments:

Please address the following comments:

Abstract: Add 95% confidence intervals to the estimates

Response: These have been added. 

Tables/Figures: Add footnotes describing the abbreviations used in the data displays (e.g., hospital name)

Response: We have added ‘CI confidence interval’ and hospital names to the footnotes of all relevant figures and tables in the manuscript and supplementary appendix. 

Page 22, line 32: Typo - eliminate word “around”

Response: ‘around’ deleted

Page 22, line 37: Type – increasing or decreasing?

Response: ‘increasing’ deleted

---

## [Editor Report · Decision Letter 2]

27 Sep 2024

Changes in urgent and emergency care activity associated with COVID-19 lockdowns in a sub-region in the East of England: interrupted times series analyses .

PONE-D-23-42956R2

Dear Dr. Bachmann,

We’re pleased to inform you that your manuscript has been judged scientifically suitable for publication and will be formally accepted for publication once it meets all outstanding technical requirements.

Kind regards,

Niklas Bobrovitz

Academic Editor

PLOS ONE

---

## [Editor Report · Acceptance letter]

15 Oct 2024

PONE-D-23-42956R2 

PLOS ONE

Dear Dr. Bachmann, 

I'm pleased to inform you that your manuscript has been deemed suitable for publication in PLOS ONE. Congratulations! Your manuscript is now being handed over to our production team.

Kind regards, 

on behalf of

Dr. Niklas Bobrovitz 

Academic Editor

PLOS ONE